**Data Availability Statement:** Data underlying the study is available at DOI: 10.6084/m9.figshare.13102913.

# The cost-effectiveness of a program to reduce intrapartum and neonatal mortality in a referral hospital in Ghana

Stephanie Bogdewic[1☉], Rohit Ramaswamy[1☉]*, David M. Goodman[2‡], Emmanuel K. Srofenyoh[3], Sebnem Ucer[4], Medge D. Owen[5‡]

**1** Gillings School of Global Public Health, University of North Carolina, Chapel Hill, North Carolina, United States of America, **2** Winnie Palmer Hospital for Women and Babies, Orlando, Florida, United States of America, **3** Greater Accra Regional Hospital, Ghana Health Service, Accra, Ghana, **4** Kybele Inc, Lewisville, North Carolina, United States of America, **5** Department of Anesthesiology, Wake Forest School of Medicine, Winston-Salem, North Carolina, United States of America

☉ These authors contributed equally to this work.
‡ These authors also contributed equally to this work.
* ramaswam@email.unc.edu

## Abstract

### Objective

To evaluate the cost-effectiveness of a program intended to reduce intrapartum and neonatal mortality in Accra, Ghana.

### Design

Quasi-experimental, time-sequence intervention, retrospective cost-effectiveness analysis.

### Methods

A program integrating leadership development, clinical skills and quality improvement training was piloted at the Greater Accra Regional Hospital from 2013 to 2016. The number of intrapartum and neonatal deaths prevented were estimated using the hospital's 2012 stillbirth and neonatal mortality rates as a steady-state assumption. The cost-effectiveness of the intervention was calculated as cost per disability-adjusted life year (DALY) averted. In order to test the assumptions included in this analysis, it was subjected to probabilistic and one-way sensitivity analyses.

### Main outcome measures

Incremental cost-effectiveness ratio (ICER), which measures the cost per disability-adjusted life-year averted by the intervention compared to status quo.

### Results

From 2012 to 2016, there were 45,495 births at the Greater Accra Regional Hospital, of whom 5,734 were admitted to the newborn intensive care unit. The budget for the systems strengthening program was US $1,716,976. Based on program estimates, 307 (±82)

**Funding:** This work was part of the 'Making Every Baby Count Initiative' awarded to PATH by the Children's Investment Fund Foundation. Kybele received a subaward (CIF.1838-01-705622-SUB) to improve maternal and newborn care capacity in regional hospitals in Ghana. RR, ES and MO were funded through this award. The funders had no role in study design, data collection and analysis, decision to publish, or preparation of the manuscript.

**Competing interests:** The authors have declared that no competing interests exist.

neonatal deaths and 84 (±35) stillbirths were prevented, amounting to 12,342 DALYs averted. The systems strengthening intervention was found to be highly cost effective with an ICER of US $139 (±$44), an amount significantly lower than the established threshold of cost-effectiveness of the per capita gross domestic product, which averaged US $1,649 between 2012–2016. The results were found to be sensitive to the following parameters: DALYs averted, number of neonatal deaths, and number of stillbirths.

## Conclusion

An integrated approach to system strengthening in referral hospitals has the potential to reduce neonatal and intrapartum mortality in low resource settings and is likely to be cost-effective. Sustained change can be achieved by building organizational capacity through leadership and clinical training.

## Introduction

Preventing neonatal death is a priority for national governments and the global health community. In 2015, the global rate of neonatal deaths, defined as a death occurring within the first 27 days of an infant's life, was 18.6 per 1,000 live births [1]. According to current estimates, roughly 2.5 million neonatal deaths occurred globally in 2018, with approximately 40% in sub-Saharan Africa (SSA) [2]. The neonatal mortality rate (NMR) remains significantly higher in SSA (27.8 deaths per 1,000 live births), compared to high-income countries (2.78) [1]. In Ghana, NMR was estimated at 25.3 deaths per 1,000 live births; while this number has decreased between 1990 and 2015 (-3.17) [1], the Ghanaian government estimates that neonatal deaths account for nearly 40% of deaths among children under the age of five [3]. If current trends persist, more than 60 countries, accounting for 80% of neonatal deaths globally, will miss the Sustainable Development Goal (SDG) target of 12 neonatal deaths per 1,000 live births by 2030 [4].

Evidence-based training packages to improve neonatal care in low and middle-income countries (LMICs) such as the WHO Essential Newborn Care Course [5], Helping Babies Breathe [6], Kangaroo Mother Care [7], and the S.T.A.B.L.E program [8] have had limited success; as training alone may be insufficient to achieve long-term, sustainable behavior change and outcomes [9]. Clinical training does not overcome broad operational and leadership gaps in resource-constricted environments [9]. Quality improvement (QI) approaches have been used to improve neonatal and obstetric care in low-resource settings. For example, Franco and Marquez's review of QI work in 12 countries found better compliance with best practices and improved outcomes, within 6 to 12 months of initial implementation [10]. Specifically, the review analyzed a variety of content areas (i.e., maternal, newborn and child health, family planning, malaria, tuberculosis, and HIV/AIDS) in different settings, and authors used indicators associated with quality and relevant client knowledge, behavior, or health status to measure improvement; each indicator was analyzed as an individual time-series chart. Overall, 87.4% of charts improved by at least 80%, and 72.6% of charts improved by 90% [10]. In Ghana, "Project Fives Alive!" implemented QI approaches to scale a set of interventions that were aimed at reducing preventable deaths among children under the age of five. Greater intensity, or the number of changes implemented, was associated with a 0.4% decrease in under-five mortality (P <0.05) [11]. However, many QI programs focus on a limited range of

problems and leave unaddressed gaps. QI researchers have stated that many QI initiatives are time limited, small- scale projects that do not building capacity for continual change and improvement of multiple processes of care [12].

One of the most challenging environments in LMICs for the provision of maternal and newborn care is within referral and teaching hospitals, where mortality rates are high due to late referrals, overcrowding, staff shortages and limitations of advanced technology and commodities for treating sick mothers and neonates [9]. In these facilities, there are no simple solutions, and multiple interventions to strengthen both clinical care and service delivery are required. The authors developed such an approach for regional referral hospitals to address clinical knowledge deficits, leadership and operational gaps simultaneously across disciplines to improve maternal and neonatal outcomes as part of the Making Every Baby Count Initiative (MEBCI) [9, 13]. The intervention focused primarily on aspects of triage, neonatal resuscitation, and labor and delivery management supplemented by QI capacity building and leadership training. Findings on the outcomes and cost-effectiveness of a similar intervention addressing maternal mortality in a regional hospital in Ghana, are reported elsewhere [9, 14].

In addition to filling existing gaps in approaches to improve newborn care, the intervention analyzed here seeks to offer a cost-effective approach combining training, quality improvement, and leadership development. To date, a number of hospital-based interventions targeting improved newborn care in sub-Saharan Africa have been assessed for cost-effectiveness, yet a gap remains in the current literature around costs associated with systems-strengthening activities. Rather, much of the existing research focuses on service-based or provider-level interventions [15–18]. For example, cost analysis of an aligned clinician-training intervention, part of MEBCI implementation in other regions, found that the cost per capita was 984 United States Dollars (US $)–a significant overall cost [19]. This purpose of this paper is to evaluate the cost-effectiveness of a hospital-based systems strengthening program intended to reduce intrapartum and neonatal mortality in Accra, Ghana.

## Methods

### Description of program activities

Kybele and GHS maintain a long-standing relationship, beginning in 2007 with efforts to reduce maternal, fetal, and neonatal mortality through a quality improvement initiative [20, 21]. A previous analysis determined the cost-effectiveness of Kybele-GHS efforts to reduce maternal and fetal mortality between 2007 and 2011 [14]. The current initiative was undertaken at the Greater Accra Regional Hospital (GARH) as a component of MEBCI, a five-year collaboration between the Ghana Health Service (GHS), Kybele and PATH to improve newborn outcomes through government engagement and provider training across four regions of Ghana.

MEBCI sought to strengthen the skills of healthcare personnel to improve newborn care through a multifaceted approach that included training in Helping Babies Breathe [6], Essential Care for Every Baby [22] and infection prevention; accessibility of resuscitation devices; and advocacy to enhance stakeholder relationships and national leadership [21]. For regional hospitals, these interventions were intended to be reinforced through a broader set of systems strengthening activities, but the scope of work was altered by the funder and the intervention analyzed in this study, and described in detail elsewhere [9], was only implemented at the GARH. Among Ghana's regional referral hospitals, GARH has the highest volume obstetric unit with 70% of deliveries comprised of high-risk antenatal or peri-partum referrals [21]. The neonatal intensive care unit (NICU) was enlarged in 2013; however, funds were not available to significantly increase the clinical workforce during the intervention period.

**Table 1. Training modules conducted at Greater Accra Regional Hospital, 2013–2016.**

| Training module | Sessions conducted | Number of Participants |
|---|---|---|
| Labor and Delivery | 3 | 61 |
| Triage Training | 8 | 63 |
| Neonatal Resuscitation | 9 | 120 |
| Continuous Positive Air Pressure (CPAP) Training | 8 | 42 |
| Quality Improvement | 4 | 37 |
| NICU Hand Washing | 1 | 19 |
| Individual Coaching Sessions | 9 | 30 |
| Leadership Charter | 1 | 25 |
| Emotional Intelligence/Leadership Styles | 3 | 22 |
| Accountability | 3 | 22 |
| Leadership Ambassador Training | 1 | 6 |
| Clinical Champion Training | 1 | 9 |
| Compassionate Care | 6 | 140 |
| **TOTAL** | **57** | **596** |

To support the capacity building approach at GARH, expert practitioners in obstetrics, neonatology, anesthesiology, midwifery and quality improvement from the United States and England made tri-annual visits to Ghana. During these visits, volunteer practitioners provided coaching and mentorship to GARH providers which were focused on introducing new clinical skill sets and optimizing care delivery. The principles of Kybele's partnership model, and the robust drivers of the successful partnership, have been described elsewhere [23]. Motivated frontline healthcare workers were selected to serve as clinical champions to facilitate learning for colleagues and to monitor data. Key performance gaps in each clinical care area were identified through the analysis of processes and baseline outcome data. Training modules were developed to address content-specific gaps in each clinical area, while foundational training and coaching was provided on QI and leadership to enable GARH staff to test, adapt and implement solutions (Table 1) that included workflow redesign, institution of compassionate care and improved communication [9, 24, 25].

An electronic database (Microsoft Access, Version 15.0, Redmond, WA) and local data sources were used to collect information on neonatal outcomes and their drivers. The Institutional Review Board (IRB) approval was granted by Wake Forest University and the GHS to conduct this work, and informed consent was waived as part of the approval.

## Description of program costs

Program costs were collected in real time to account for multiple sources and types of costs incurred; researchers kept a detailed budget during the intervention, and costs were reported in US $. This analysis includes costs incurred in 2012, as they were directly related to planning for the implementation of this intervention. All costs were standardized to 2019 US $.

1. *Grant funding*: The grant funding for the MEBCI project primarily offset the travel costs and professional time of a subset of experts, administration, accommodation, training, incidentals and other direct costs. Participants paid for travel insurance, medicines, visa fees and other expenses not reimbursed by the funder.

2. *Volunteered professional time*: In addition to the grant funding the project team received, other participants provided voluntary, uncompensated services. Twenty-seven health professionals traveled to Ghana and worked for 484 days at GARH during the project; fifteen

of whom (55%) returned multiple times. The 2016 U.S. Department of Labor, National Occupational Employment, and Wage Estimates, Role and Occupation Code was used to determine the cost of volunteer contributions, based on 10-hour work days [26]: $1,100 /day for physicians and quality improvement specialists, and $500/day for midwife or nurse practitioner. Since these are intrinsically variable, we tested our calculations with a 25% variability during the sensitivity analysis, per similar analyses [14].

3. *Services provided by Ghana Health Service*: The GHS provided transportation for the volunteers, and invested in the construction cost of a new triage pavilion at the GARH.

4. *Other donations*: Duke Medical Center, Access Bank, East Meets West Foundation and volunteers donated medical devices such as, fetal dopplers, continuous positive airway pressure devices, blood pressure monitors, airway resuscitation bags and neonatal hats.

The program costs incurred by Kybele, along with external aid, accounted for 62% of the total cost, and the estimated cost of professional time accounted for 26%. The calculated program cost was US $1,590,276. Given that this analysis is from the perspective of the non-governmental organization, it does not consider any changes to the costs of delivering care.

The calculated program costs have been adjusted for inflation and standardized to 2019 USD using the Consumer Price Index (CPI). The CPI for all items was used to adjust the program related costs, the physician services component of medical care category was used to adjust the costs of professional value time, and the medical care commodities component to adjust the costs of medical equipment and supplies [27]. Additionally, a majority of the funding for this project came from international sources; thus, this funding is not subject to purchasing-power parity (PPP) adjustments. All costs incurred in Ghanaian currency, including compensated time from Ghanaian providers and costs covered by GHS, were adjusted to account for PPP by using PPP exchange rates that factor price levels in different countries based on a standard basket of goods and services [28]. While the price levels for health-related services may differ from the basket used to calculate the PPP exchange rate, the fact that GHS investments covered both capital expenditures and provider time allows an adequate approximation. The inflation-adjusted total program cost is US $1,716,976 (Table 2); operational and infrastructure program costs are also presented (Table 3).

**Table 2. Kybele-MEBCI program cost in US $.**

|  | Reported program cost (US $) | Inflation-adjusted Costs (US $) |
|---|---|---|
| **Kybele**[a] | 981,577.62 | 1,059,899.06 |
| **Participant**[b] | 31,938.20 | 35,042.51 |
| **Value-time**[c] | 417,200.00 | 449,346.75 |
| **GHS**[d] | 92,229.72 | 99,744.44 |
| **Others**[e] | 67,330.43 | 72,943.36 |
| **Total** | **1,590,276** | **1,716,976** |

[a] Funder costs included travel and administrative expenses, accommodations, training, and other direct costs

[b] Participant costs included travel expenses of participants

[c] Value time indicates the calculated value time for volunteers

[d] GHS costs included food expenses, and the construction of a new triage pavilion

[e] Other, third-party costs included donated medical equipment; costs associated with the renovation of the NICU (4,980.28 US $) were omitted from the present analysis, but these costs would amount to 0.3% of total costs and would not impact findings or conclusions.

**Table 3. Reported operational and infrastructure program costs in US $.**

| | Operational Costs (US $)[a] | Infrastructure Costs (US $)[b] | Total Costs (US $) |
|---|---|---|---|
| May 2012—July 2012 | $49,507 | | $49,507 |
| Sept 2012—Nov 2012 | $19,340 | | $19,340 |
| Jan 2013—March 2013 | $75,876 | | $75,876 |
| May 2013—July 2013 | $38,717 | $300 | $39,017 |
| Sep 2013—Nov 2013 | $97,617 | | $97,617 |
| Dec 2013—Feb 2014 | $153,784 | $42,960 | $196,744 |
| Mar 2014—May 2014 | $156,542 | $3,000 | $159,542 |
| Jun 2014—Aug 2014 | $58,431 | | $58,431 |
| Sep 2014—Nov 2014 | $112,782 | | $112,782 |
| Dec 2014—Feb 2015 | $148,858 | $82,918 | $231,776 |
| Mar 2015—May 2015 | $167,297 | $3,125 | $170,422 |
| Jun 2015—Aug 2015 | $186,313 | | $186,313 |
| Sep 2015—Nov 2015 | $83,959 | $2,500 | $86,459 |
| Dec 2015—Feb 2016 | $104,373 | $2,077 | $106,450 |
| **Total** | **$1,453,396** | **$136,880** | **$1,590,276** |

[a] Operational costs include travel expenses, administrative expenses, value time, training, incidentals, and other direct costs; it is important to note that operational costs associated with value time account for salaries from international staff, and these costs would be reduced were the program to be run by GHS

[b] Infrastructure costs include equipment, donated items, and construction

## Estimating disability-adjusted life years (DALY) for neonatal and fetal death in Ghana

The disability-adjusted life year (DALY) is the most commonly used summary measure to quantify the burden of disease within a given population in LMIC [28–31]. The DALY metric relies on the assumption that the most appropriate measure of the effects of a chronic illness is time, both time lost due to premature death and time spent disabled by disease [31]. Therefore, DALYs are calculated by summing the adjusted number of years lived with disability (YLDs) and the number of years of life lost due to premature mortality (YLL) [32].

YLD = Number of cases x duration until remission or death x disability weight

YLL = Number of deaths x life expectancy at the age of death

DALY = YLD + YLL

The Global Burden of Disease (GBD) project provides guidance on methods considered best practice for calculating DALYs. The major philosophical and methodological aspects of the DALY calculation have been described and debated [32–34]. The recent GBD does not discount future DALYs [35], which removes the assumption that current years of life lost are valued at a higher rate than future years of life lost. The current GBD does not apply age weighting, or the concept that the value of years lost varies with age. Historically, WHO has used both age weighting and discounting future DALYs when calculating YLL [36, 37], and this analysis takes both methodologies into account. Researchers have described methods associated with discounting DALYs elsewhere [38]. This study uses standard values for age weighting and discounting [39].

Although typical DALY calculations rely on years of life lost to both death and disability, it is not common for cost-effectiveness analyses of neonatal health interventions in LMICs to include YLD estimates [38–40]. Due to the challenges associated with accurately estimating the long-term impact of disabilities, the current study relies on YLL to estimate DALYs [39]. The discount rate (r) was set at 3% according to the WHO standard for economic evaluation

of health interventions in LMICs [37]. The YLL due to premature death were calculated using the average of Ghana-specific life expectancy at birth for male and female, as local life expectancy is recommended as a good approximation of life expectancy (L) [38]. Early neonatal deaths account for 76% of neonatal deaths globally and were assumed to be the age at the event for the calculation of YLL (a) [41].

$$\text{YLL} = \frac{KCe^{ra}}{(r+\beta)^2}\left[e^{-(r+\beta)(L+a)}[-(r+\beta)(L+a)-1] - e^{-(r+\beta)a}[-(r+\beta)a-1]\right] + \frac{1-K}{r}(1-e^{-rL})$$

a = age at death, in years
r = discount rate
$\beta$ = age-weighting constant
K = age-weighting modulation factor
C = constant from age-weighting function
L = standard life expectancy at birth, in years

Additionally, researchers are engaged in discussion around the inclusion of stillbirths in DALY calculations [42, 43]. The current study does not attempt to assign a value to life lost in utero prior to the onset of labor; the authors are conscious that the loss of a fetus places a great cost on families. In the 2013 Global Health Estimates, the WHO published recommendations around the inclusion of stillbirths as years of life lost and based the value on life expectancy at birth [44]. The present study included stillbirths in the base and sensitivity analysis, but considered only fresh stillbirths due to intrapartum complications, that is fetuses that arrived at the hospital with a heart beat but were born dead.

## Estimating number of deaths and DALYs averted through the partnership

To estimate the number of deaths and DALYs averted due to the partnership, this study compares the number of neonatal deaths avoided to a "no-intervention" counterfactual. This counterfactual was not observed, but rather estimated as a steady-state scenario that would have occurred had the intervention not been implemented. The quasi-experimental pre- and post-program evaluation, which was used to inform the estimation of the counterfactual, relied on data collected annually from non-random sites. In this method, a baseline NMR has been used to predict the number of neonatal deaths that would have occurred if the intervention had not taken place. An electronic project database was developed to collect project outcome indicators; primary data sources for outcome indicators included the Delivery Register in the Maternity Ward and the Newborn Admission and Discharge Register in the NICU, which are routinely collected following a patient encounter. The NMR from GARH in 2012 has been used as baseline NMR, with training starting January 2013. Thus, to calculate the estimated number of neonatal deaths under the steady-state assumption, the number of reported babies delivered at GARH in a given year was multiplied by the hospital's 2012 NMR of 3.11%. Compared to this estimated baseline, the authors determined that any reduction in neonatal deaths would be seen as an improvement. However, this approach of estimating neonatal deaths averted through a steady state assumption is likely to be an over-estimate of the number of neonatal deaths averted by the intervention. Additionally, it is difficult to attribute causality to the intervention, as NMR may be impacted by existing demand- and supply-side factors. Similarly, the 2012 stillbirth rate (SBR) was used to make steady-state assumptions in order to estimate the number of stillbirth deaths averted in subsequent years. Following calculation of the deaths prevented, the DALYs averted were calculated using the same assumptions discussed above.

## Assessing the cost-effectiveness of treatment

The incremental cost-effectiveness ratio (ICER) is a metric used to determine the cost-effectiveness of a program or interventions. For this study, an ICER shows the program cost-effectiveness as measured in estimated DALYs averted by the program compared with a null hypothesis of no change.

$$ICER = \frac{Cost_{QI} - Cost_{Null}}{DALY_{QI} - DALY_{Null}}$$

Estimates of costs, health effects, and ICERs provide clear guidance to policymakers only when an explicit threshold standard or threshold has been specified among other conditions [43]. In the absence of an explicit standard or threshold by policymakers, it would be difficult to make an objective recommendation. The ceiling ratio ($\lambda$), or decision rule, is an important component of cost-effectiveness analysis (CEA) and represents the decision makers' valuation of a unit of health gain or the relative value against which the acceptability of ICERs is judged [45]. While explicit cost-effectiveness thresholds are available for US and UK, the selection of $\lambda$ for interventions affecting LMICs has been left to the discretion of the analyst [45, 46]. Within LMIC settings, researchers most commonly use a cost-effectiveness threshold based on per capita gross domestic product (GDP). This approach has been promoted by the WHO-CHOICE project to define cost-effectiveness of an intervention [47, 48]. If the cost of averting one DALY is less than three times the national annual GDP per capita then an intervention is deemed cost-effective, and if it is less than once the country-specific GDP per capita it is considered highly cost-effective [41, 49]. Cost-effectiveness research throughout sub-Saharan Africa has widely utilized a threshold determined by GDP [50–53].

The league table approach, derived from the work of World Bank, recommends US $150 per DALY as 'attractive' cost-effectiveness, US $25 per DALY as 'highly attractive' for low-income countries and US $500 and US $100 per DALY, respectively, for middle-income countries [45, 46]. Each of the approaches has advantages and disadvantages; for the purpose of this study, the authors present results using multiple approaches to determine the cost-effectiveness of the systems strengthening intervention.

## Sensitivity analysis

To test assumptions made in the analysis, the authors subjected the data to a probabilistic sensitivity analysis using Monte Carlo simulations, run in Crystal Ball (Oracle, Redwood Shores, CA) as an add-in program to Microsoft Excel (Microsoft, Redmond, WA). All assumptions were varied simultaneously according to pre-specified distributions. The distributions were assigned according to the inherent characteristics of each parameter and according to accepted conventions and based on a similar CEA conducted for maternal mortality [14]. In order to calculate DALYs using discounting, the following parameters were applied (Table 4): the assumptions for average age and life expectancy were distributed uniformly around high and low estimates; the value of professional time was varied at 25%; and the number of neonatal deaths and stillbirths were varied around a normal distribution. Using the Monte Carlo simulation, the researchers modeled uncertainty in the program estimates.

## Results

### Estimating DALYs averted through the partnership

The baseline number of babies delivered at GARH and the outcome data are presented in Table 5. Over the course of the intervention, NMR decreased by 19.4% in 5 years. Compared

**Table 4. Sensitivity analysis parameters.**

| Parameter | Low | Mid | High | Distribution of variation |
|---|---|---|---|---|
| Average age | 0.01728 | 0.0192 | 0.02112 | Uniform |
| Life expectancy | 56.25 | 62.5 | 68.75 | Uniform |
| Stillbirth life expectancy | 56.25 | 62.5 | 68.75 | Uniform |
| K (age weighting) | - - - - | 1 | - - - - | Used in discounting formula |
| C (constant) | - - - - | 0.1658 | - - - - | |
| r (discount) | - - - - | 0.03 | - - - - | |
| B (age weight function) | - - - - - | 0.04 | - - - - | |
| Professional time | US $ 337,010.06 | US $ 449,346.75 | US $ 561,683.44 | Uniform |
| 2013–2016 Observed neonatal deaths | -1.96* SE | Observed yearly value | + 1.96* SE | Normal |
| 2013–2016 Estimated number of neonatal deaths assuming steady state | -1.96* SE | Observed yearly value | + 1.96* SE | Normal |
| 2013–2016 Observed Stillbirths | -1.96* SE | Observed yearly value | + 1.96* SE | Normal |
| 2013–2016 Estimated number of neonatal deaths using Ghana's NMR | -1.96* SE | Observed yearly value | + 1.96* SE | Normal |

SE- Standard error for sample; NMR- Neonatal Mortality Rate

to the steady-state NMR, 307 neonatal deaths were averted during the intervention years (Fig 1). Over the course of MEBCI, Ghana experienced an annualized rate of change of NMR of 1.26% [54]. Thus, this improvement would explain 2.5% of the neonatal deaths averted. The intrapartum SBR decreased 30.6%, and an estimated 84 stillbirths were prevented during the intervention years (Fig 2). As discussed in the methods section, while decreases in observed NMR and SBR are considered to be an improvement, it is difficult to contribute these changes entirely to MEBCI, as there could be demand-side and supply-side factors impacting these data.

The DALY per one neonatal death averted was 31.57 (62.5 undiscounted) (Table 6), and the DALY per one stillborn averted was 31.55 (62.5 undiscounted) as the average age of a stillborn is zero. The steady-state assumption estimates that 307 neonatal deaths were averted during the program, leading to 9,692 DALYs avoided. If the 84 prevented stillbirths are also included, an additional 2,650 DALYs were averted, for a total of 12,342 DALYs averted during the program years.

**Table 5. Baseline data for the partnership.**

| Parameter | 2012 | 2013 | 2014 | 2015 | 2016 |
|---|---|---|---|---|---|
| Total number of babies delivered at GARH | 11338 | 7549 | 9672 | 8807 | 8129 |
| Observed number of neonatal deaths | 353 | 179 | 165 | 208 | 204 |
| Observed NMR (per 1000 live births) | 31.12 | 23.78 | 17.10 | 23.59 | 25.08 |
| Estimated number of neonatal deaths, based on steady-state NMR assumptions | 353 | 235 | 301 | 274 | 253 |
| Neonatal deaths prevented, due to steady-state NMR assumptions | 0 | 56 | 136 | 66 | 49 |
| Observed number of intrapartum stillbirths | 135 | 77 | 96 | 83 | 67 |
| Observed SBR (per 1,000 live births) | 11.91 | 10.15 | 9.91 | 9.41 | 8.27 |
| Estimated number of stillbirths, based on steady-state SBR assumptions | 135 | 90 | 115 | 105 | 97 |
| Stillbirths prevented, based on steady-state SBR assumptions | 0 | 13 | 19 | 22 | 30 |

NMR -Neonatal Mortality Rate; SBR- Stillbirth Rate. The table shows the neonatal deaths and stillbirths prevented assuming steady-state, based on data from institutional reports; most data were compared from two sources and the most conservative estimates were used. The baseline rate for the steady-state assumption of neonatal deaths averted and SBR is 2012.

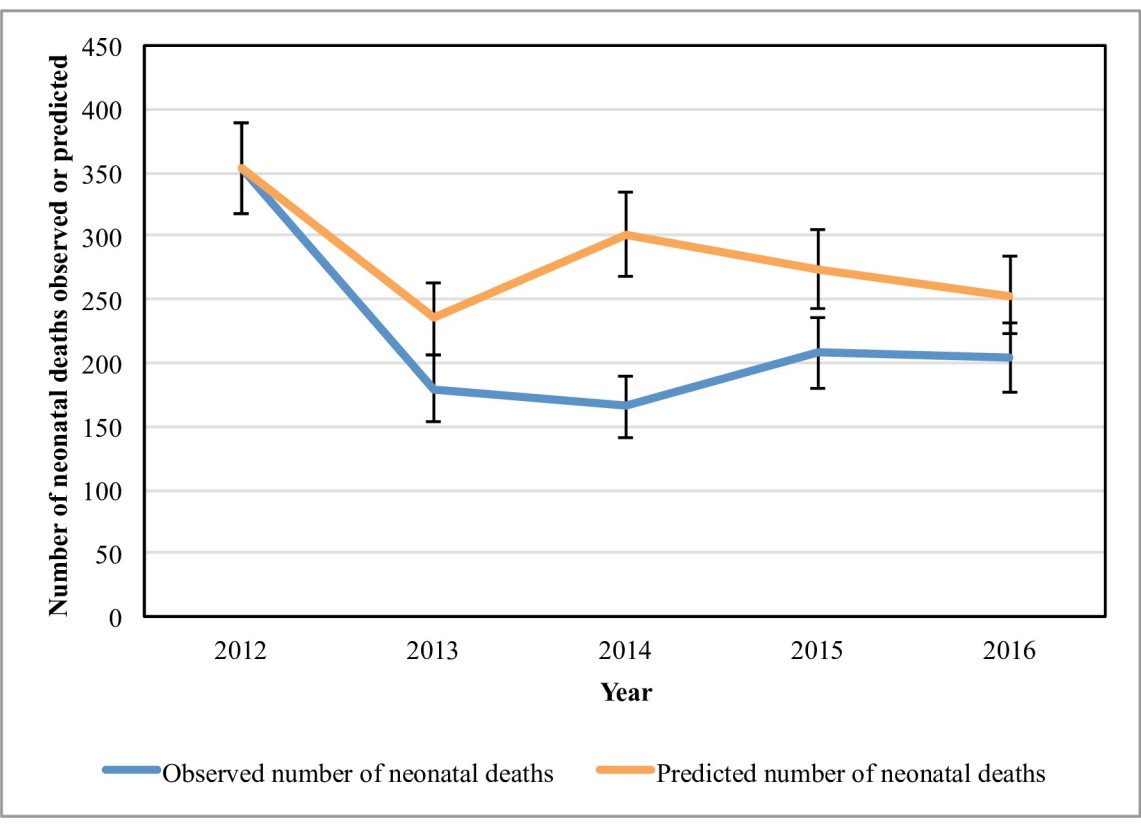

**Fig 1. Neonatal deaths predicted to have occurred, by steady-state assumption.**

## Cost-effectiveness analysis

ICER represents the extra units of outcome achieved per extra dollar spent. Under the steady-state assumption, compared to status quo, the intervention costs an additional US $1,716,976 but also results in 12,342 DALYs averted for an incremental cost-effectiveness ratio of US $139/DALY averted. Under the commonly used threshold based on per capita GDP (US $1,649, Ghana's average GDP for 2012–2016), it is likely that the intervention for reducing neonatal deaths is highly cost-effective. Additionally, in the league table approach, the World Bank recommended a threshold of US $100 for 'highly attractive' and US $500 for 'attractive' cost-effectiveness for a middle-income country like Ghana [45]. With an ICER of US $139/DALY, it is likely that the systems strengthening intervention for reducing neonatal deaths is considered 'attractive' for cost-effectiveness.

## Sensitivity analysis: Considering the changing environment

As the current study presents a retrospective analysis, it was important to identify and consider the assumptions that were made around the unobserved counterfactual (Table 4). In order to account for a degree of uncertainty in this analysis, performance was varied as a normal distribution, centered on an observed mean. The distribution sampling was then repeated, using the identified parameters, in Monte Carlo to obtain credibility estimates (Table 7). The tornado charts for the analysis (Fig 3) shows the 5th and 95th percentile estimates for each assumption included in the sensitivity analysis, as well as how that variation affects the estimates of the ICER. Variables at the top of the tornado chart indicate those that have the largest impact on

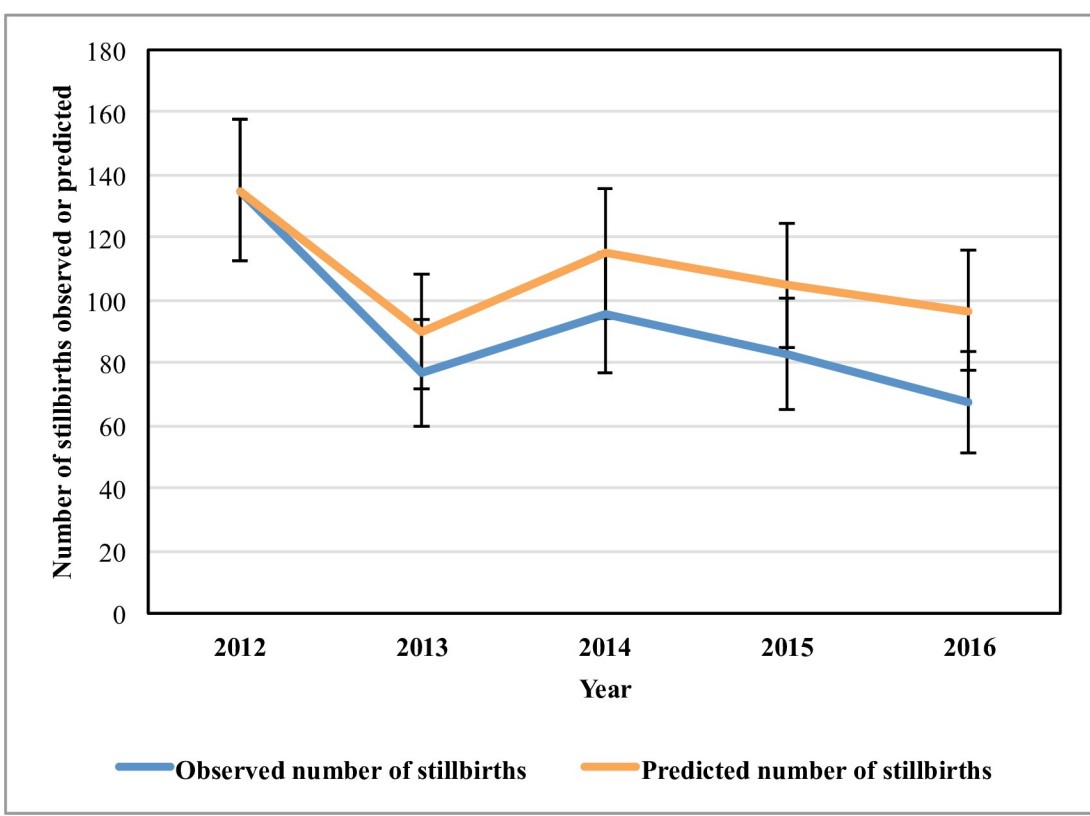

**Fig 2. Stillbirths predicted to have occurred, by steady-state assumption.**

the variability of the ICER estimates. The most significant assumption is the estimated number of neonatal deaths in 2015, accounting for 12.2% of variation in the ICER estimate; the 2012 NMR was used as a baseline estimate for this analysis. The next most significant assumptions were the estimated number of neonatal deaths in 2014 and 2016, each accounting for 11.9% of the variation. Variables contributing to ≥4% of the variation in ICER estimates have been

**Table 6. Years of life lost (YLL) for each neonatal death using different social value choices.**

| Variables | YLL [0.03,1][a] | YLL [0,1][b] | YLL [0,0][c] | YLL [0.03,0][d] |
|---|---|---|---|---|
| Age weighting modulation factor (K) | 1 | 1 | 1 | 1 |
| Age weighting constant (β) | 0.04 | 0.04 | 0 | 0 |
| Discount rate (r) | 0.03 | 0 | 0 | 0.03 |
| Age at event (a) | 0.0192 | 0.0192 | 0.0192 | 0.0192 |
| Constant from the age weighting function (C) | 0.1658 | 0.1658 | 0 | 0 |
| Standard life expectancy at birth (L) | 62.5 | 62.5 | 62.5 | 62.5 |
| Epsilon | 2.72 | 2.72 | 2.72 | 2.72 |
| **YLL per each neonatal death averted** | **31.57** | **73.92** | **62.5** | **28.2** |

Table 6 provides the variables used to calculate YLL, adjusting for weighting and discounting, as well as the final YLL calculation under each social value choice.

[a] Discounted YLL calculation adjusting for age weighting

[b] Discounted YLL calculation, not adjusting for age weighting

[c] Non-discounted YLL calculation, not adjusted for age weighting

[d] Non-discounted YLL calculation, adjusted for age weighting

**Table 7. Sensitivity analysis using steady-state rate.**

| Parameter | DALYs Base Case | 95% CI | ICER Base Case | 95% CI |
|---|---|---|---|---|
| Discounting neonatal deaths | 9,691 | 5,503–13,836 | 177.17 | 122.75–312.55 |
| Discounting neonatal deaths and stillbirths | 12,342 | 7,722–16,885 | 139.23 | 100.80–223.17 |
| Undiscounted neonatal deaths | 19,188 | 10,713–27,692 | 89.48 | 61.44–159.90 |
| Undiscounted neonatal deaths and stillbirths | 24,419 | 15,203–33,744 | 70.31 | 50.66–112.93 |

shown in the tornado chart, and the top variables contributing to maximum variation are all assumptions made from the institutional data. All institutional data were compared from two sources and the most conservative estimates were used in the analysis.

## Discussion

### Main findings

This study presents the cost-effectiveness of a hospital-based systems strengthening intervention to improve newborn health using prospectively collected data. Analysis shows that with US $1,716,976 invested, including nearly US $500,000 worth of donated professional time, the partnership prevented an estimated 307 neonatal deaths and 84 intrapartum stillbirths,

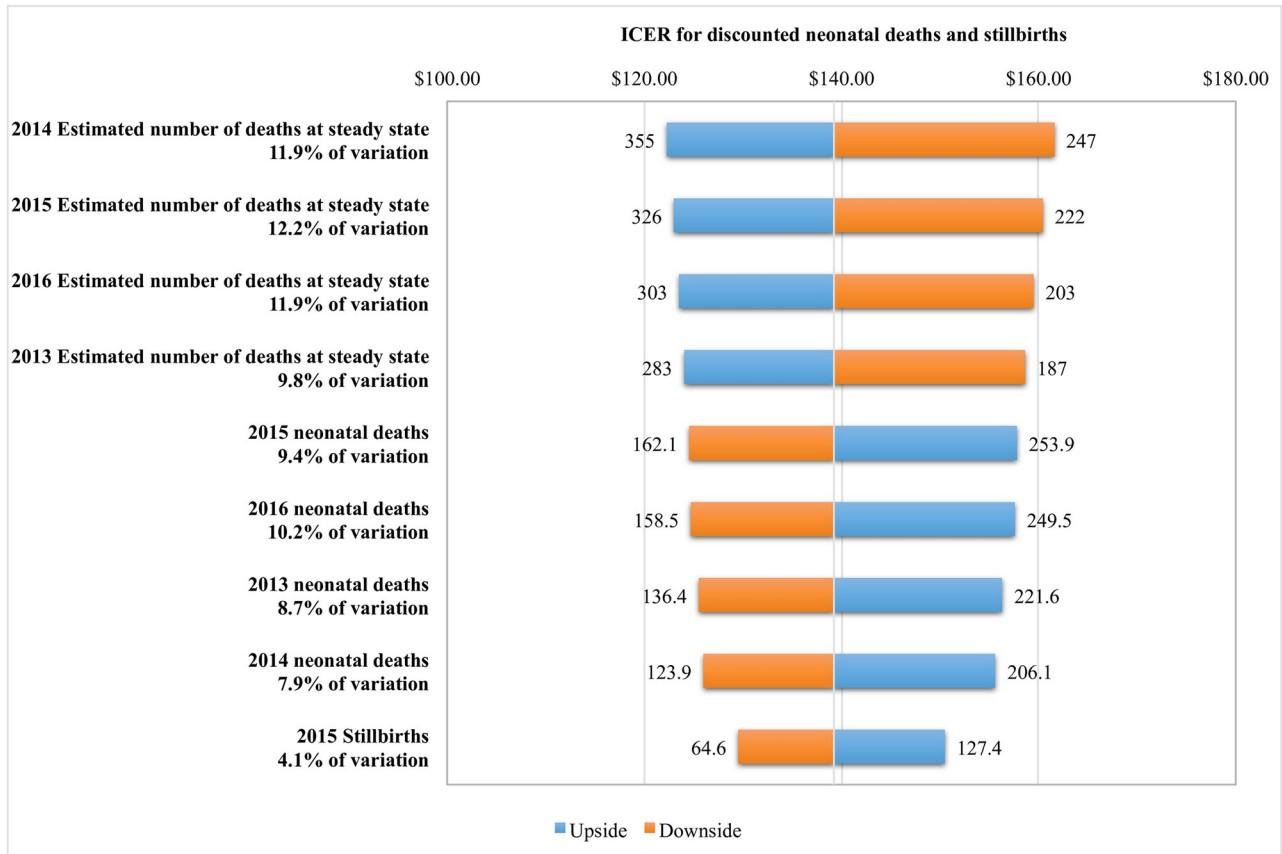

**Fig 3. Tornado chart of ICER for discounted calculations of neonatal deaths and stillbirths estimated from steady-state assumption showing variables contributing ≥4% of the variation in the estimates.** ICER—incremental cost-effectiveness ratio; upside—assumption makes the intervention more cost effective; downside—assumption makes the intervention less cost effective.

averting 12,342 discounted DALYs. The sensitivity analysis provides 95% confidence that the DALYs averted ranged from 7,722 to 16,885 under the steady-state assumption. With an ICER of US $139/DALY, the intervention can be regarded highly cost effective.

It is difficult to attribute causality in studies such as this due to the inability to systematically control the intervention. With the more modest intervention package undertaken in the other four regional referral hospitals as part of the MEBCI, the newborn referral rate increased from 6.7 referrals per 1,000 live births to 10.8 [55]; stillbirths and mortality rates were unchanged over the same time course [56]. Although increases in referrals were only measured in the other four regions, it is likely that GARH also experienced an increase in referrals of complicated cases. Under the assumption that GARH saw increased referrals, it is likely that the "no intervention" counterfactual experiences increased NMR because of increasing prevalence of complicated cases. This potential impact of increased prevalence of high-risk cases would lead to an under-estimation of the mortality that occurred in the "no intervention" counterfactual. However, it is also possible that improvements could have been made at GARH without the influence of this program due to secular trends, but we cannot accurately predict the effect those improvements would have made. Such improvements that may have happened at GARH without the intervention would lead to an over-estimation of the impact of this analysis. It is important to note that although GARH providers may have attended opportunistic trainings during implementation, there were no additional activities taking place at GARH during the course of the intervention that were directly related to systematically improving neonatal care.

As discussed in a similar analysis, it is uncommon for NGOs to report cost-effectiveness related to the use of QI interventions to reduce neonatal and fetal mortality [14]; however, this study followed Consolidated Health Economic Evaluation Reporting Standards for reporting cost-effectiveness to the extent possible [57]. The hospital-based systems strengthening intervention was intended to advance clinical care quality and efficiency, which may result in reduced costs. The current study is a single study-based estimate that represents the cost of long-term coaching, leadership development, and quality improvement initiatives from which other NGOs may benefit. While the heterogeneity of local context makes it difficult to generalize these findings beyond GARH, other NGOs can use a similar approach for analyzing costs.

In addition to providing findings that may benefit NGOs and funders, this analysis can provide insight for the GHS in making decisions about ongoing activities to improve neonatal care. By strengthening capacity to improve neonatal care at the provider and systems levels, the intervention can have lasting impact. Learning from this implementation has been used to inform the Obstetric Triage Implementation Package [58], as well as the scale up of the systems-strengthening model in three other facilities, led by GHS. Although it was appropriate to present the cost-effectiveness findings from the NGO perspective, overall findings from this analysis are pertinent for local policy- and decision-makers in that implementing the model with local experts would reduce the overall cost.

The current analysis does not address the fact that GARH serves as a referral hospital, and outreach to district hospitals, other educational programs and governmental policy improvements have all occurred during the program years. The impacts associated with these efforts were not addressed in this study, and they may serve to overestimate the cost-benefit analysis conducted here. Additional analysis is needed to understand the impact of these types of activities.

## Strengths and limitations

This study has several strengths. First, it provides an applied example of a retrospective analysis of an intervention, rather than relying on prospectively developed hypothetical constructs.

This program was built on an established relationship between Kybele and the GHS utilizing quality improvement to enhance maternal care at GARH [14]. The strength of the partnership led to funding to extend the integrated approach to improve neonatal outcomes. NGOs working to improve neonatal and fetal health do not typically report cost-effectiveness, and this study may encourage other NGOs to report cost-effectiveness and to promote transparency to donors and host countries. The intervention supported the development of leadership and QI skills among providers at GARH with the goal of impacting sustained improvements in care. Finally, partnership with a local bank led to an investment to renovate the NICU, demonstrating the value the local community places on improving newborn health, which is also likely to influence sustainability.

However, the study also has weaknesses that are important to discuss. This study does not provide a societal, or Ghana's Ministry of Health, perspective. Additionally, the project was largely conducted using grant funding from an international organization, and foreign experts' donated time. It is unlikely for many tertiary hospitals in LMICs to have access to skilled professionals for multidisciplinary mentoring, although this should be recognized as a gap in building high-level maternal and newborn best practices in global health contexts. However, ongoing initiatives, as a result of collaboration between the researchers and GHS, seek to address this gap and support local capacity. Third, this study suffers from incomplete health care and case fatality rate data as is seen in other studies from LMICs [59]. The probabilistic sensitivity analysis and the tornado charts do lend support that the analysis relied on data that was, at times, incomplete. Fourth, systems strengthening interventions are generally time intensive; thus, programs such as this may be difficult to replicate in other settings due to limitations in staffing and funding. This study was conducted in a single hospital, and it is likely that the exact number of neonatal deaths averted cannot be known. Additionally, the NICU at GARH did not meet international standards, although GARH is a moderately well-staffed and equipped hospital. Similar considerations in other healthcare systems could limit the validity of cost-effectiveness system strengthening interventions, as improvements also depend on staffing and infrastructural capabilities of hospital settings.

The researchers found that training in QI, leadership, and clinical care across disciplines along a continuum of care resulted in numerous, sustained process improvements. As local providers and clinical champions become better equipped in QI methodology and leadership through change, the cost of similar capacity building programs, or healthcare in general, could decrease. The long-term sustainability and scalability of this approach at the regional hospital level requires additional consideration.

## Conclusion

As national governments in LMICs prioritize strategies to reduce neonatal mortality to achieve SDG targets, an integrated hospital-based approach to systems strengthening holds promise. In this study, the system strengthening approach integrating quality improvement, leadership development and clinical skills enhancement through an established partnership likely reduced neonatal mortality in a regional referral hospital in Ghana. It is estimated that 307 (±82) neonatal deaths and 84 (±35) stillbirths were prevented amounting to 12,342 DALYs averted. According to the threshold of estimating cost effectiveness per capita GDP, the intervention was found to be highly cost-effective with an ICER of $139 (±$44), which is significantly less than the average Ghanaian per capita GDP of US $1,649. Study findings support the hypothesis that using an integrated approach to systems strengthening in referral hospitals, developed through the partnership of local and visiting healthcare providers, holds promise for producing cost-effective outcomes for infants in Ghana and other low resource settings.

## Author Contributions

**Conceptualization:** Medge D. Owen.

**Data curation:** Sebnem Ucer.

**Formal analysis:** Stephanie Bogdewic.

**Funding acquisition:** Medge D. Owen.

**Methodology:** Rohit Ramaswamy, David M. Goodman.

**Supervision:** Rohit Ramaswamy, Medge D. Owen.

**Validation:** Emmanuel K. Srofenyoh.

**Writing – original draft:** Stephanie Bogdewic.

**Writing – review & editing:** Stephanie Bogdewic, Rohit Ramaswamy.

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
