## [Decision Letter · Decision Letter 0]

21 Aug 2020

PONE-D-20-10170

The cost-effectiveness of a systems strengthening program to reduce intrapartum and neonatal mortality in a referral hospital in Ghana

PLOS ONE

Dear Dr. Ramaswamy,

Thank you for submitting your manuscript to PLOS ONE. After careful consideration, we feel that it has merit but does not fully meet PLOS ONE’s publication criteria as it currently stands. Therefore, we invite you to submit a revised version of the manuscript that addresses the points raised during the review process.

The reviewers have recommended major revisions to your paper. Please address their comments especially the issues of affordability and sustainability in the study context and similar settings.

We look forward to receiving your revised manuscript.

Kind regards,

Tanya Doherty, PhD

Academic Editor

PLOS ONE

Journal Requirements:

2) Thank you for stating in your ethics statement that "the study qualified for expedited review and met the

criteria to waive informed consent". In your ethics statement in the manuscript and in the online submission form, please clarify whether informed consent was actually waived by the ethics board as part of your IRB approval.

3)  We noticed you have some minor occurrence of overlapping text with the following previous publication(s), which needs to be addressed:

https://journals.plos.org/plosone/article?id=10.1371%2Fjournal.pone.0180929

In your revision please quote or rephrase any duplicated text outside the methods section. Further consideration is dependent on these concerns being addressed.

4) We note that you have included the phrase “data not shown” in your manuscript. Unfortunately, this does not meet our data sharing requirements. PLOS does not permit references to inaccessible data. We require that authors provide all relevant data within the paper, Supporting Information files, or in an acceptable, public repository. Please add a citation to support this phrase or upload the data that corresponds with these findings to a stable repository (such as Figshare or Dryad) and provide and URLs, DOIs, or accession numbers that may be used to access these data. Or, if the data are not a core part of the research being presented in your study, we ask that you remove the phrase that refers to these data.

5) PLOS requires an ORCID iD for the corresponding author in Editorial Manager on papers submitted after December 6th, 2016. Please ensure that you have an ORCID iD and that it is validated in Editorial Manager. To do this, go to ‘Update my Information’ (in the upper left-hand corner of the main menu), and click on the Fetch/Validate link next to the ORCID field. This will take you to the ORCID site and allow you to create a new iD or authenticate a pre-existing iD in Editorial Manager. Please see the following video for instructions on linking an ORCID iD to your Editorial Manager account: https://www.youtube.com/watch?v=_xcclfuvtxQ

6) Please amend either the abstract on the online submission form (via Edit Submission) or the abstract in the manuscript so that they are identical.

7) Please include captions for your Supporting Information files at the end of your manuscript, and update any in-text citations to match accordingly. Please see our Supporting Information guidelines for more information: http://journals.plos.org/plosone/s/supporting-information.

Reviewers' comments:

Reviewer's Responses to Questions

**Comments to the Author**

1. Is the manuscript technically sound, and do the data support the conclusions?

Reviewer #1: Partly

Reviewer #2: Yes

2. Has the statistical analysis been performed appropriately and rigorously? 

Reviewer #1: I Don't Know

Reviewer #2: Yes

3. Have the authors made all data underlying the findings in their manuscript fully available?

Reviewer #1: Yes

Reviewer #2: Yes

4. Is the manuscript presented in an intelligible fashion and written in standard English?

Reviewer #1: Yes

Reviewer #2: Yes

5. Review Comments to the Author

Reviewer #1: Review of: The cost-effectiveness of a systems strengthening program to reduce intrapartum and

neonatal mortality in a referral hospital in Ghana

Major comments:

This is an interesting article which is worthy of publication after some editing to ensure that it does not over-claim what was done or what was achieved.

The first major point is that in my opinion the term “systems strengthening” is an over-claim and should be removed from the title and throughout. This is not really what I would call a “systems strengthening” intervention. Judging by table 1 it was a training intervention largely or solely delivered by foreign experts. It only involved one tertiary referral hospital (not the health system as a whole), and involved 27 health professionals travelling to Ghana from the USA / UK and working for 484 days at GARH during the project. This is a rather top-down approach. No mention is made of training local staff to deliver the training to their colleagues to make this sustainable, or of making the training more widely available throughout the health system. Mention is made of a few other hospitals, but no mention is made of primary care or other parts of the health system. Why was a “train the trainers” approach not taken, which would have made the training sessions much cheaper, more sustainable, and more cost-effective? $1100 per day is probably several orders of magnitude greater than doctors in Ghana are paid – as such there is great scope for making this much more cost-effective by training trainers locally and reducing the massive foreign input, which as the authors mention is not scalable or replicable throughout Africa.

The second major point is that little information is provided on other ongoing interventions or changes which could also have had an impact on neonatal mortality. The authors state “there could be demand-side and supply-side factors impacting these data.” – are they aware of any other specific interventions or factors taking place in their study site in 2012-2016 which could have impacted on neonatal or perinatal mortality? Were there changes in health insurance payments for example? The discussion states “partnership with a local bank led to an investment to renovate the NICU” – was this taken into account in the costs? This could also explain part of the reduction in mortality…

The discussion states “It is likely, therefore, that the “no intervention” counterfactual could have had increasing mortality rates as a consequence of increasing prevalence of high-risk cases” – but is there any data to support this, on prevalence of high-risk cases in each year? If so this data should be presented. Certainly overall numbers were falling so this would not support this argument. The total number of babies delivered reduced from 11338 in 2012 to 8129 in 2016 (a 28% reduction). Why is this? Could this also explain the reduction in mortality, as the hospital would have been less stretched and there would have been more resources available per baby (assuming human resources and other factors remained constant)?

Thirdly, the authors claim this was a “quality improvement” intervention but very little information is provided (apart from table 1) of what was done to improve quality, and whether anything other than mortality was measured to see whether quality did indeed improve after the numerous training sessions. For example were any audits done of neonatal resuscitation to see whether this was being better implemented after the training? What key performance gaps were identified? Were these measured again after the training to see if they had improved? Did clinicians perform maternal and perinatal death surveillance and reviews? If so which avoidable factors were identified, what recommendations were made to address them, and were these recommendations implemented?

Fourthly, the cost-effectiveness calculation is impressive but parts of it are not intelligible to a non-health economist. I would recommend making this clearer to those of us who are not experts in health economics. For example, the YLL equation does not include an explanation as to what the letters stand for. Figure 3 and Table 5 need further explanation to make them clearer to non-specialists.

Regarding cost-effectiveness, I think the data actually makes it fairly clear that the model of sending out many foreign experts to conduct training (which is often done as part of health partnerships) is probably NOT cost-effective by African standards and that more cost-effective models are needed. The cost-effectiveness threshold stated is theoretical but has it ever been used by any country in Africa? It seems to me more akin to thresholds used in high-income countries. Just one simple example: a dose of surfactant costs about $90 which can save the live of a premature baby with respiratory distress. According to the authors’ calculations this would translate to about $3 per DALY (assuming a discounted life expectancy of about 30 years). Yet this is deemed to be too expensive in most if not all African countries, and as a result surfactant is either not available at all, or maybe only in private hospitals which are not affordable to the majority. Was surfactant available at the Ghanaian hospital, and if not, how much would it have cost, and how many lives could have been saved in comparison with the reported intervention? It seems to me highly improbable that any African country would be willing to spend $139 per DALY on any intervention. The discussion should recognise this and should also consider and discuss ways in which the intervention could have been made more cost-effective and replicable. There should be a greater focus on lessons learned and which parts of this intervention might be relevant to other settings and might be scalable.

Minor correction needed:

“NMR decreased by 19.4% in 5 years and neonatal deaths reduced from 353 to 204 per 1,000 live births” - According to table 4 these are absolute numbers, not a rate per 1000 live births. The rate per 1000 live births is reported as 31.12 and 25.08.

Table 5: to a non health economist it is not clear what the figures mean in square brackets in the title row of the table, such as [0.3, 1]. Please explain this in the legend.

p21: “Fourth, systems strengthening interventions are generally time intensive; thus,

programs such as this may be difficult to replicable” change to “Fourth, systems strengthening interventions are generally time intensive; thus, programs such as this may be difficult to replicate”

Reviewer #2: The authors tackle an important pubic health area- interventions to reduce neonatal mortality in SSA, specifically Ghana. This study aims to examine the costs & cost-effectiveness of a systems strengthening programme, under the MEBCI initiative, compared to the status quo. This programme was delivered at a regional referral level hospital in Ghana. It specifically sought to enhance clinical knowledge & skills (e.g. triage, neonatal resuscitation), labour and management delivery practices as well as narrow operational gaps. Results indicate that this programme is associated with an estimated prevention of 307 neonatal deaths and 84 stillbirths, amounting to 12,342 DALYs averted. The authors find that this programme is highly cost-effective. Findings were generally robust to alternate model specifications.

The authors have previously reported on outcomes from a similar intervention targeting maternal mortality (Goodman et al., 2017).

I have applied the Consolidated Health Economic Evaluation Reporting Standards (CHEERS) checklist for assessing the quality of this study.

Introduction:

- The authors provide a clear description of the issue at hand and cite key literature.

- Could the authors elaborate on this sentence on page 5 “However, many QI programs focus on a limited range of problems and leave unaddressed gaps.” It would also be worth highlighting some the QI interventions aimed at reducing neonatal mortality in SSA, with regards to effectiveness and costs. See for example (Cavicchiolo, M.E, et al., 2016. Reduced neonatal mortality in a regional hospital in Mozambique linked to a Quality Improvement intervention. BMC pregnancy and childbirth, 16(1), p.366.)

- Some specifics on how does this study expand on/fill specific gaps in the neonatal mortality intervention lit?

- Authors should include specific objectives of this economic evaluation.

- In the final paragraph, it would be important to highlight the relevance of this CEA for health policy makers and programme planners.

Methods:

- Could the authors highlight what part of the current programme was informed by the previous programme evaluated by this group (Goodman et al., 2017)? It is unclear how different or similar the previous programme is to the current programme. A diagram/appendix highlighting what each component targeted is needed, particularly the health systems strengthening components (e.g. under QI- what aspects did this cover?)

- In the abstract, this study is described as “quasi-experimental and time-sequence”- could the authors elaborate on this in the methods. What about this makes it a quasi-experimental /time-sequence and not simply a pre-post evaluation?

- Perspective & costs estimates- a stronger justification on why a perspective of the NGO was taken and why costs are not estimated using Ghanaian rates- particularly volunteer HCWs? Surely findings have major implications for the Ghanaian MOH. For sustainability reasons, it would be beneficial to the MOH on what are the implications for scale-up of this programme/integration this within routine services. I would recommend that the authors present primary findings taking into account Ghanaian salary rates. An appendix could be inserted for costs estimates calculated using USA HCW salary rates.

- Table 2 should be included in the results and expanded on.

- The authors provide a detailed explanation on how they estimated DALYs.

- Some elaboration is needed on which specific data sources were used to extract key outcome data. Where any of these routine data sources?

- The authors should consider some assessment of affordability. Part of the modelling should consider what the estimated financial cost would be if the programme was run by the MOH under routine set-up, using various scenarios.

Results

- A table comparing financial versus economic costs by the various stages (Annualized set-up and 1-year implementation costs) would be important for policy makers. Include both fixed and variable costs (i.e. capital, overheads, training, transport).

Discussion

- The discussion should consider implication for the Ghanaian government. In the introduction the authors note the importance of this intervention for Ghana, but the opening statements in this section refer to impact of this programme for the donor only.

- How does the cost per DALYs averted in this programme compare to others economic evaluations of programmes aimed at reducing neo-natal mortality more broadly in this region?

- It is unclear why the authors indicate this study “does not lend itself to policy decisions”. As this programme was part of the broader government MEBCI, surely it has implications for the Ghanaian MOH ? Given current global funding issues, some analysis should be conducted to discuss implications around affordability and sustainability.

- Some discussion on the generalisability of the findings and key biases should be discussed.

- The authors highlight key limitations- some information on how this would have impacted (i.e. over-/under-estimated) the overall outcome (cost per DALYs averted) should be discussed.

6. PLOS authors have the option to publish the peer review history of their article (what does this mean?). If published, this will include your full peer review and any attached files.

Reviewer #1: **Yes: **Merlin Willcox

Reviewer #2: No

---

## [Author Response · Author response to Decision Letter 0]

23 Oct 2020

Please view attached file for detailed responses to reviewer comments

---

## [Editor Report · Decision Letter 1]

28 Oct 2020

The cost-effectiveness of a program to reduce intrapartum and neonatal mortality in a referral hospital in Ghana

PONE-D-20-10170R1

Dear Dr. Ramaswamy,

We’re pleased to inform you that your manuscript has been judged scientifically suitable for publication and will be formally accepted for publication once it meets all outstanding technical requirements.

Kind regards,

Tanya Doherty, PhD

Academic Editor

PLOS ONE

Additional Editor Comments (optional):

Thank you for your comprehensive response to the reviewer comments and the revisions made to the manuscript. It is now deemed suitable to be accepted.
---

## [Editor Report · Acceptance letter]

5 Nov 2020

PONE-D-20-10170R1 

The cost-effectiveness of a program to reduce intrapartum and neonatal mortality in a referral hospital in Ghana 

Dear Dr. Ramaswamy:

I'm pleased to inform you that your manuscript has been deemed suitable for publication in PLOS ONE. Congratulations! Your manuscript is now with our production department. 

Kind regards, 

on behalf of

Professor Tanya Doherty 

Academic Editor

PLOS ONE